# Insights on the Choice and Preparation of the Donor Nerve in Corneal Neurotization for Neurotrophic Keratopathy: A Narrative Review

**DOI:** 10.3390/jcm13082268

**Published:** 2024-04-14

**Authors:** Pietro Luciano Serra, Giuseppe Giannaccare, Alberto Cuccu, Federico Bolognesi, Federico Biglioli, Marco Marcasciano, Filippo Tarabbia, Domenico Pagliara, Andrea Figus, Filippo Boriani

**Affiliations:** 1Plastic Surgery Unit, Department of Medical, Surgical and Experimental Sciences, University of Sassari, Sassari University Hospital Trust, 07100 Sassari, Italy; pietro_serra@hotmail.it; 2Plastic Surgery and Microsurgery Unit, Department of Surgical Sciences, Faculty of Medicine and Surgery, University Hospital “Duilio Casula”, University of Cagliari, 09124 Cagliari, Italy; andreafigus@hotmail.com (A.F.); borianifilippo@gmail.com (F.B.); 3Eye Clinic, Department of Surgical Sciences, University of Cagliari, 09124 Cagliari, Italy; cuccualberto@yahoo.it; 4Department of Maxillo-Facial Surgery, Santi Paolo e Carlo Hospital, University of Milan, 20122 Milan, Italy; federico.bolognesi2@gmail.com (F.B.); federico.biglioli@unimi.it (F.B.); filippo.tarabbia@asst-santipaolocarlo.it (F.T.); 5Plastic and Reconstructive Surgery Unit, Division of Plastic and Reconstructive Surgery, Department of Surgery, China Medical University Hospital, Taichung, Taiwan, Magna Graecia University of Catanzaro, 88100 Catanzaro, Italy; dott.marcomarcasciano@gmail.com; 6Unit of Plastic and Reconstructive Surgery, Department of Experimental and Clinical Medicine, Magna Graecia University, 88100 Catanzaro, Italy; 7Plastic-Reconstructive and Lymphedema Microsurgery Center, Mater Olbia Hospital, 07026 Olbia, Italy; domenico.pagliara@materolbia.com

**Keywords:** corneal, neurotization, nerve, coaptation, graft

## Abstract

The article introduces neurotrophic keratopathy (NK), a condition resulting from corneal denervation due to various causes of trigeminal nerve dysfunctions. Surgical techniques for corneal neurotization (CN) have evolved, aiming to restore corneal sensitivity. Initially proposed in 1972, modern approaches offer less invasive options. CN can be performed through a direct approach (DCN) directly suturing a sensitive nerve to the affected cornea or indirectly (ICN) through a nerve auto/allograft. Surgical success relies on meticulous donor nerve selection and preparation, often involving multidisciplinary teams. A PubMed research and review of the relevant literature was conducted regarding the surgical approach, emphasizing surgical techniques and the choice of the donor nerve. The latter considers factors like sensory integrity and proximity to the cornea. The most used are the contralateral or ipsilateral supratrochlear (STN), and the supraorbital (SON) and great auricular (GAN) nerves. Regarding the choice of grafts, the most used in the literature are the sural (SN), the lateral antebrachial cutaneous nerve (LABCN), and the GAN nerves. Another promising option is represented by allografts (acellularized nerves from cadavers). The significance of sensory recovery and factors influencing surgical outcomes, including nerve caliber matching and axonal regeneration, are discussed. Future directions emphasize less invasive techniques and the potential of acellular nerve allografts. In conclusion, CN represents a promising avenue in the treatment of NK, offering tailored approaches based on patient history and surgical expertise, with new emerging techniques warranting further exploration through basic science refinements and clinical trials.

## 1. Introduction

Neurotrophic keratopathy (NK) is a pathologic cascade deriving from the denervation of the cornea [1], due to various conditions affecting the function of the trigeminal nerve [2]. The effects of the absent nerve supply to the cornea are an impaired sensitive and trophic functioning, which causes damage to the corneal epithelium [3].

Surgery of the peripheral nervous system is a relatively novel but rapidly evolving field which is merging important and intriguing elements of microsurgery, basic neuroscience knowledge, and tissue engineering. The term “neurotization” has ancient origins, as it was coined by the Arabic scientist Avicenna in the eleventh century to define regeneration of sectioned nerves by directly connecting their stumps by sutures. Nowadays, this definition has been replaced by the term “nerve coaptation”, whereas the modern meaning of neurotization refers to the connection of nerve fibers to a target organ, either motor or sensory.

Corneal neurotization (CN) represents the procedure in which the trigeminal nerve is surgically recovered as the carrier of sensitive function to the cornea. It was initially proposed without success by Samii in 1972 [4] with a very invasive approach as it entailed a craniotomy and an intracranial coaptation between the occipital nerve and the proximal ophthalmic nerve with a sural nerve graft. An alternative and less invasive surgical option was then proposed by Terzis in 2009 [5]. This technique consisted of a direct surgical corneal neurotization (SCN) in patients suffering from facial nerve palsy in addition to ipsilateral trigeminal nerve disfunction and corneal denervation. This procedure represented a revolution in the surgical treatment of NK, since it opened the way to a variety of possible options that were developed in the following years. CN is defined as “direct” if the trigeminal donor nerve is directly sutured or fixed to the cornea, while it is defined as “indirect” if a nerve graft is employed [6].

The selection and preparation of the donor nerve, the harvesting of the autologous nerve graft, and the choice of the allogenic, xenogeneic, or heterologous nerve graft are relevant parts of this approach and pertain to technical knowledge that is outside the borders of historic and traditional ophthalmology. Hence, this type of operation must be multidisciplinary, with the involvement of ophthalmologists and experts on reconstructive microsurgical techniques on the peripheral nerve system.

Therefore, the aim of this review is to provide the surgeon with the necessary knowledge of the broad technical armamentarium that should guide the surgical approach of CN.

## 2. Methods

PubMed research was conducted using the terms “corneal neurotization”, “corneal neurotization”, and “neurotrophic keratopathy”. The most relevant English articles published from 2015 to date were selected. A total of 35 articles were included (Table 1), and based on these, a review of the latest surgical techniques, focusing on the choice and preparation of the donor nerve and the technique used, was conducted. The most relevant techniques are summarized in Table 2.

## 3. Results

The cornea receives sensory innervation through the ciliary nerves of the ophthalmic nerve (V1), the first branch of the trigeminal nerve (cranial nerve V). Innervation is crucial for corneal homeostasis, playing a role in tear production and epithelial regeneration [7]. Neurotrophic keratopathy is a degenerative disease caused by progressive trigeminal damage that results in corneal hypoesthesia. It has a prevalence of 1.6–11/10,000, and its most frequent cause is post-herpetic keratitis [8,9]. Other causes of trigeminal damage at different levels include diabetes, dry eye, tumors, surgical trauma, and refractive surgery [7,10,11,12,13]. The repair of peripheral nerve injuries has historically been performed by coapting the proximal side of the severed nerve with the distal one, either directly or through nerve grafting, to avoid tension at the suture site. Coaptation initiates a process of axonal regeneration proceeding from the distal side of the proximally injured nerve to the stump of the distal nerve or directly into the target tissue. This process begins with Wallerian degeneration and continues with Schwann cell dedifferentiation and proliferation, aided by tissue macrophages and inflammatory cells [14]. Young age and a shorter distance between severed or injured nerve endings (gap length) are associated with better outcomes [15,16,17,18]. It has been demonstrated that sensory end organs can survive denervation for an extended period; therefore, surgical repairs can occur several years after the initial injury and still yield satisfactory results [17,19,20,21]. Both nerve autografts and allografts have demonstrated comparable outcomes [18]. Injuries such as neuropraxia and axonotmesis have a good prognosis [20], contrary to neurotmesis, which has a poor prognosis [22]. The first sensory improvements are observed 6 months postoperatively and can continue up to 3 years [16,17]. Nerve endings can be connected through an end-to-end (E-E), end-to-side (E-S), or side-to-side (S-S) neurorrhaphy [23,24,25,26]. The E-E technique has shown overall better functional recovery, associated with increased nerve fiber count, area, and density [24]. In this publication, Rönkkö and coworkers set up an experimental study on 80 rats, in which a proximal nerve lesion (section of the common peroneal nerve) was treated with E-E, E-S, and S-S. Each repair corresponded to a study group, and, in addition, an unrepaired group and a sham un-sectioned subset were included. Outcome measuring considered the peroneal functional index (PFI) as the primary endpoint—that is, an indication of the recovery of the leg function specifically related to the common peroneal nerve, obtained through walking track analysis. Moreover, histomorphometric post-mortem variables were taken into account. In terms of PFI, at the longest follow up (26 weeks), the E-E group outperformed E-S and S-S subsets, with a relevant statistical significance (*p* < 0.001). With regard to histomorphometry, in the same timeframe, the fiber count, total fiber area, fiber density and percentage of the fiber area related to E-E outperformed E-S (all *p* < 0.02) and S-S (all *p* < 0.001). The final choice of surgical technique depends on the relative sizes of the donor and graft nerves since similar caliber is necessary for good alignment in the E-E approach. The E-S coaptation may be more suitable when there are significant differences in nerve caliber, the donor nerve is critical for function, or the proximal nerve ending is unavailable [27,28].

Several techniques for corneal neurotization have been described. A sensory donor nerve can be directly or indirectly transferred to the affected cornea through the interposition of an autologous graft or allograft nerve [29] that is decellularized before use. The choice depends on factors such as the availability of the sensory donor site, the size of the donor nerve, the surgeon’s experience, and the distance between the donor nerve and the cornea [9]. The donor nerve in most techniques utilized belongs to the complex trigeminal arborization. However, in some descriptions, the great auricular nerve is employed as the donor source of sensitive innervation [30,31]. The great auricular nerve is a cutaneous sensory nerve originating from the second and third cervical spinal nerves (C2–C3) of the cervical plexus.

Terzis et al. [5] were the first to describe the technique for DCN, using a coronal incision and isolating the contralateral supraorbital (SON) or supratrochlear nerve (STN), which were then tunneled through the bridge of the nose to the affected eye. They were extricated through an incision in the upper eyelid fold, then tunneled through the upper fornix onto the ocular surface. Incisions in the bulbar conjunctiva allowed the nerve to be placed between the sclera and Tenon’s capsule. Finally, the branches were sutured, and the conjunctiva closed over them. Alternatively, the infraorbital nerve (ION) can be used [29,32], but this requires a lower orbitotomy and unroofing of the infraorbital canal to expose the nerve. In this case, an E-S neurorrhaphy is typically performed [29]. Nowadays, the approach to the SON and STN is typically through the upper eyelid fold or a subciliary incision. It can also be performed minimally invasively through an endoscopic approach with minimal scars [33]. Improvements are visible from 3 months postoperatively up to 3 years in terms of recovery of corneal sensitivity, healing of the NK, and reduction in both corneal neovascularization and opacity. Furthermore, the denervation time does not influence the success rate of the procedure [5,29,32,34,35,36,37].

Regarding ICN, the most commonly used nerve grafts are the sural nerve (SN) [30,38,39,40,41,42], the great auricular (GAN) [31,43], and the lateral antebrachial cutaneous nerve (LABCN) [44]. Historically, in plastic surgery, the SN has been the preferred choice for nerve grafts due to its easy harvesting, long length (up to 40 cm), and minimal sensory deficits [45,46,47,48,49]. Its mean caliber is 3.6 mm, originates from the union of the medial and lateral sural cutaneous nerves at the distal third of the gastrocnemius (running superficially to it), and then courses posteriorly to the lateral malleolus. It innervates the posterolateral skin of the leg and the lateral region of the foot, heel, and ankle. It can be easily harvested with an open technique [50,51]. A recent systematic review [52] analyzed 240 patients undergoing SN harvesting, reporting a loss of sensitivity in 87.2% of cases, residual pain in 25.6%, cold sensitivity in 22.2%, and functional impairment in 10%. A valid alternative to the SN is represented by the LABCN. It is an exclusively sensory branch of the musculocutaneous nerve and provides sensitivity to the anterolateral surface of the forearm. It can be harvested through a single longitudinal incision with its 4 terminal branches for a total length of approximately 12 cm. The sensory deficit resulting from its harvesting is minimal and in a non-critical anatomical area. Furthermore, the LABCN presents a similar caliber of the SON (1.3–1.8 mm vs. 1.1–1.7 mm, respectively) and a similar number of fascicles [44]. Another nerve graft option is the GAN. It is a cutaneous nerve originating from the middle cervical loop of the cervical plexus and consists of fibers from C2 and C3. It innervates the skin of the parotid and mastoid regions, part of the auricle, and the parotid fascia. Its harvesting as a graft is also associated with low morbidity. Its emergence point is estimated to be 6.5 cm from the mastoid and 1 cm from the external jugular vein [53]. It can be harvested through a single 3 cm incision in the neck at the midportion of the nerve with a length of approximately 7 cm and an average diameter of 2.6 mm [43]. An advantage of using this nerve is a shorter surgical time because only one surgical field is prepared. Within the field of ICN, another emerging possibility is the use of allografts [29,54]. For instance, when there is a shortage of autologous graft nerves, grafts from cadavers can be used. Patients receiving an allograft must be immunosuppressed for at least 2 years until the donor nerve is recellularized with the recipient’s Schwann cells. Indeed, there is an immunogenic response triggered by the Schwann cells present in the allograft [55]. It is also possible to decellularize allografts through enzymatic degradation and irradiation to remove the cells and immunogenic material from the nerve. This technique preserves the extracellular matrix and nerve architecture, facilitating axonal regeneration and eliminating the need for immunosuppression [56]. The decellularized allograft serves as a scaffold to be repopulated by the host’s axons and Schwann cells. The use of allografts allows for benefiting from the same characteristics as autografts, with advantages such as less donor site morbidity, unlimited availability, and reduced surgical time. However, their use still carries a high cost [57]. Furthermore, it is possible to avoid the coronal incision approach, with fewer scars, reduced risk of alopecia, damage to the frontal branch of the facial nerve, and postoperative hematoma. According to the study by Leyngold et al. [29], the recovery of the donor nerve dermatome is faster compared to the coronal or endoscopic approach. Overall, allografts have shown to have a functional success rate equal to autografts both in sensory and motor reinnervation in gaps of up to 7 cm with a wide variety of nerve diameters [58,59,60].

Similar to the choice of the donor nerve, a combination of preoperative and intraoperative factors, as well as the surgeon’s skills and preferences, should be considered. Before surgery, it is necessary to assess the sensitivity of the donor area, immediately ruling out any already deficient donor nerves. Patients whose cause of NK is central will likely have both V1 and V2 affected. In this case, the donor nerve must be contralateral or located elsewhere (for example, the GAN). It is crucial to minimize the graft length as much as possible to reduce the distance of the axonal regeneration and to match the caliber of the donor nerve and the graft. Moreover, it is important to cause as little damage as possible to the donor site. In accordance with these principles, the first choices are represented by the ipsilateral SON and STN [61]. In particular, the deep branch of the SON has a constant anatomy, with a good caliber and a robust structure. The second choices are represented by the contralateral SON and STN and the ipsilateral ION. Using the contralateral SON and STN, the nerve graft will not necessarily be longer, but the dissection will be simpler and less traumatic with fewer comorbidities compared to the ION [29]. Using the ipsilateral ION, the nerve graft is shorter but requires a more complex dissection, including orbitotomy. In this case, an E-S coaptation to the nerve graft is also necessary. Figure 1 summarizes the techniques with the described variety of donor nerves and grafts. The medical literature on this topic, as is apparent from the current review, is still very explorative on the method, with a majority of case reports and only a few case series, which do not take the form of randomized controlled trials. In addition, the mentioned case series suffer from a relevant heterogeneity affecting the type of surgical technique, the methodology, and the postoperative assessment. This hinders the possibility of meta-analyzing the results by pooling cases in order to have a sufficiently large homogeneous sample.

## 4. Representative Clinical Case

A 51-year-old man came to our attention due to NK in the right eye, complaining of hyposecretive dry eye and recurrent corneal erosions. In terms of etiology, the patient suffered from an iatrogenic lesion of the right facial and trigeminal nerve following an intracranial procedure for the surgical treatment of an acoustic neurinoma in 2012. The patient underwent ICN under general anesthesia. Giannaccare and coworkers [6] have thoroughly detailed and illustrated the surgical technique adopted by this research group, with regard to both the extra-ophthalmological and the ophthalmological parts of the procedure. The SN was isolated and harvested from the left leg through 3 incisions of 2 cm each. They were then sutured with Monocryl 3/0 for the subcutaneous tissue and Monocryl 4/0 for the skin. At the level of the left eye, an incision was made at the level of the upper eyelid crease, deep through the orbicularis muscle to the orbital septum and finally to the bony rim. The SON was identified and isolated. A subcutaneous tunnel was created at the level of the nasal bridge. An incision was then made at the level of the upper eyelid crease of the right eye in its medial third. The incision was deepened to the orbicularis muscle. The SON was transposed from the left eye, through the nasal bridge, to the right eye. Neurorrhaphy was performed between the proximal stump of the SN and the SON using Nylon 10/0 suture. The skin was sutured with Prolene 6/0. A 360° conjunctival peritomy was performed, and the axonal stumps were retrieved in the subconjunctival plane using a Wright needle. The nerve was divided into three main branches that were fixed to the cornea with fibrin glue at the sclerocorneal limbus at 10, 5, and 7 o’clock. The conjunctiva was sutured with Vicryl 7/0 and tarsorrhaphy was then performed with 5/0 silk suture.

The patient has been assessed at 3 months postoperatively. The postoperative course is still short, but the clinical findings are encouraging. When the patient applies eye drops to his right eye, he perceives a referred tactile sensation in the original territory of the donor left supraorbital nerve (skin of the left forehead).

## 5. Discussion

Despite the fact that corneal neurotization is still at an embryological phase in its course, it is a promising surgical technique thanks to the advantage it provides in treating the cause of the disease rather than the final output, as most of the other options were intended. This means that it deserves attention from the scientific community. The favorable aspect of peripheral nerve repair when a sensory end-organ is involved is the fact that differently from motor nerves, a sensory target can tolerate denervation for an extended time period; thus, surgical nerve repair or neurotization can be performed several years after the initial damage, maintaining good rates of success [17,19,20,21].

In 2020, Fogagnolo and colleagues [62] compared the DCN and ICN approaches for corneal neurotization in a non-randomized multicentric interventional study, demonstrating that NK was healed in all patients regardless of the type of surgery. The mean period for partial recovery of sensation was 3.9 months; in this case series of 26 eyes, DCN demonstrated a faster recovery and higher corneal sensitivity in the early post-operative time, due to the absence of neurorrhaphy and direct nerve sprouting on corneal surface. After one year and in long-term analysis, this difference did not reach statistical significance. Therefore, in conclusion, DCN is preferable in worse cases (Mackie grade 3 with high risk of perforation) where short time of reinnervation is mandatory to save the affected eye. A variety of techniques are available for the surgeon and the decision is based on different factors, including aspects related to the donor nerve, such as sensory integrity, nerve caliber, and axon count, anatomical vicinity to the target cornea, NK severity, previous surgeries with craniotomy and high risk of SO and ST nerve damage, unilateral or bilateral cornea anesthesia, surgical accessibility, and surgeon’s expertise and predilection. Table 3 summarizes the factors and variables determining the choice of the donor nerve.

When NK is caused by a local, ocular factor, ipsilateral trigeminal innervation is likely to be intact and therefore useful for a simpler, ipsilateral direct neurotization. In case of more central damage, an ICN based on a contralateral donor nerve may likely be necessary [63]. Other relevant aspects to be considered for the choice of the donor nerve are nerve caliber (which should match with the graft), related axon count (which should be as great as possible for the benefit of the final target organ), and the distance between donor and recipient. A good correspondence between calibers of donor and graft is advantageous for coaptation [29]. In terms of donor and graft/recipient matching, the SO nerve contains approximately 6000 myelinated axons, similar to the IO nerve. Moreover, they are both closest to an ipsilateral denervated cornea, with a larger caliber and a more constant anatomical route compared to the ST nerve, which normally consists of only 2500 myelinated axons at the orbital rim [64]. Surgical incision approaching the ST and SO nerves is conducted through the upper lid crease or in the sub-brow region, while preparation of the IO nerve involves an inferior orbitotomy and unroofing of the infraorbital canal to visualize the nerve. In the case of the IO donor, an E-S nerve coaptation is selected, as a major morbidity derives following neurotmesis and denervation of the innervated territory [29]. Corneal recovery depends on the well-described process of direct sprouting of axons from the coaptation site. However, an additional phenomenon has been theorized, especially for the initial phase in which the sprouting axons have not reached the target cornea. This is a paracrine-like release of nerve factors supplied by the graft/donor nerve [5,65]. Several in vivo and clinical evidence support the axonal sprouting effect as the main role in corneal sensation recovery. Labeled corneal nerves after a procedure of corneal neurotization in a rat model show a continuous axonal connection to the donor neurons [66]. In addition, a common postoperative finding is the synesthetic perception at the contralateral donor nerve territory upon stimulation of the recovering cornea [67]. This is confirmed by magnetoencephalography, as stimulation of the recovering cornea post-neurotization evoked a cortical activation in the area connected to the contralateral donor trigeminal nerve with no stimulus at the area corresponding to the ipsilateral trigeminal nerve [19]. Based on the superior evidence of axonal sprouting, minimizing the distance between the donor nerve and the recipient cornea is a key factor in reducing the recovery time, as supported by studies on peripheral nerve reconstruction [15]. In terms of factors influencing the surgical outcome, Table 4 summarizes positive and negative prognostic factors.

Preoperative sensory evaluation work-up includes the ST nerve (provides the medial part of the forehead, nasal bridge, and upper eyelid), the SO nerve (supplies the lateral forehead, the upper eyelid, and the anterior scalp) and the IO nerve (provides the lateral aspect of the nose, the cheek, upper lip, and upper dental arch). In case none of these nerves are available, sensation supplied by the greater auricular nerve should be tested. Acellular nerve allografts, already explored in this field by Leyngold et al. [29] and Sweeney et al. [68], represent a promising tool, especially when utilized in combination with autologous nerve [69] or when enhanced with cells or noncellular factors [70,71]. Acellular nerve allografts are nerves that undergo a pre-transplantation decellularizing treatment that make them non-immunogenic, different from previously used traditional nerve allografts which were transplanted fresh and untreated, thereby warranting the use of immunosuppression, at least for a few months. After a long period in which the only available product was represented by AVANCE, our study group has now developed a new and effective decellularization method [72]. Future directions should explore the possibility to make the procedure less invasive for the patient, more comfortable for the surgeon, and more sustainable for the health system.

The medical literature is still poor of large series, since most primary studies are case reports with a single patient. Moreover, the published series are methodologically heterogeneous and recent; therefore, there is still little follow up. The largest series on cornea neurotization is the work by Fogagnolo et al. [62] who reported a case series of 25 patients and 26 eyes, comparing DCN and ICN, followed by Sweeney and coworkers [69], who reported data from 17 patients of ICN with acellular nerve allografts. Lin and colleagues [35] published a cases series of 13 patients, and the technique utilized was a direct ipsilateral supratrochlear neurotization. Leyngold and coworkers [29] published a cases series of 7 patients treated with acellular nerve allografts-based ICN, followed by Giannaccare and collaborators [61], who published about 3 patients treated with DCN.

In conclusion, NK is a challenging and sight-threatening disease, which was historically treated with palliative medical options while corneal surgery was limited only to complicated cases. In the last few years, SCN has added a potentially game-changing tool, in which the nerve supply to the cornea has to be preceded by a number of reconstructive options, translated from peripheral nerve reconstruction. The choice of the most adequate one must be customized based on the clinical history of the patient, neuroanatomy, and the surgeon’s preferences. Acellular nerve allografts represent a promising element in the armamentarium of the multidisciplinary team facing this disease and they deserve basic science refinements and future clinical trials. Due to the rarity of this condition, clinical trials often struggle with limited patient populations. The small number of patients can make it difficult to achieve sufficient statistical power and demonstrate treatment efficacy.

## Figures and Tables

**Figure 1 jcm-13-02268-f001:**
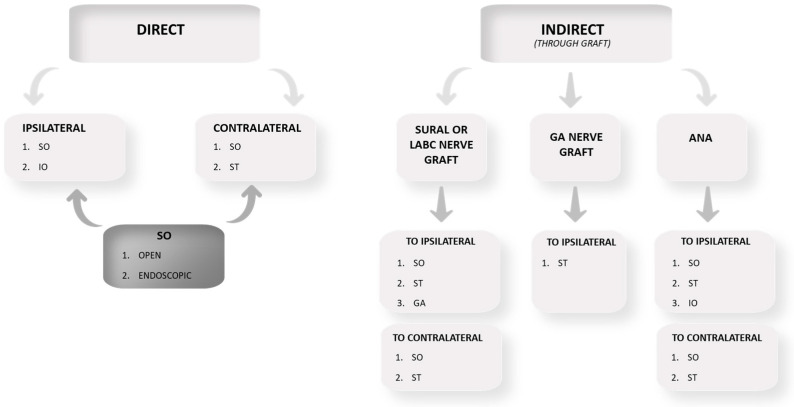
Summary of the available technique for cornea neurotization. Abbreviations: SO supraorbital, ST supratrochlear, IO infraorbital, GA great auricular, LABC lateral antebrachial cutaneous, ANA acellular nerve allograft. The length of the harvested or purchased graft for ICN should be approximately 10 cm in case of ipsilateral donor nerve or 15 cm in case of contralateral or great auricular donor nerve.

**Table 1 jcm-13-02268-t001:** PubMed research was conducted using the terms “corneal neurotization”, “corneal neurotization,” and “neurotrophic keratopathy”. The 35 most relevant English articles published from 2015 to date were included and are shown herein.

Title	Authors	Journal	Year
Corneal Neurotization With a Great Auricular Nerve Graft: Effective Reinnervation Demonstrated by In Vivo Confocal Microscopy	Benkhatar H, Levy O, Goemaere I, Borderie V, Laroche L, Bouheraoua N	*Cornea*	2018
Treatment of neurotrophic keratopathy with minimally invasive corneal neurotisation: long-term clinical outcomes and evidence of corneal reinnervation	Catapano J, Fung SSM, Halliday W, Jobst C, Cheyne D, Ho ES, Zuker RM, Borschel GH, Ali A	*Br J Ophthalmol*	2019
Corneal Neurotization via Dual Nerve Autografting	Charlson ES, Pepper JP, Kossler AL	*Ophthalmic Plast Reconstr Surg*	2022
Corneal Neurotization: A Review of Pathophysiology and Outcomes	Park JK, Charlson ES, Leyngold I, Kossler AL	*Ophthalmic Plast Reconstr Surg*	2020
Corneal Neurotization: A Meta-analysis of Outcomes and Patient Selection Factors	Swanson MA, Swanson RD, Kotha VS, Cai Y, Clark R, Jin A, Kumar AR, Davidson EH	*Ann Plast Surg*	2022
In Vivo and Ex Vivo Comprehensive Evaluation of Corneal Reinnervation in Eyes Neurotized With Contralateral Supratrochlear and Supraorbital Nerves	Giannaccare G, Bolognesi F, Biglioli F, Marchetti C, Mariani S, Weiss JS, Allevi F, Cazzola FE, Ponzin D, Lozza A, Bovone C, Scorcia V, Busin M, Campos EC	*Cornea*	2020
CORNEAL NEUROTIZATION IN A PATIENT WITH SEVERE NEUROTROPHIC KERATOPATHY. CASE REPORT	Rusňák Š, Hecová L, Štěpánek D, Sobotová M	*Cesk Slov Oftalmol*	2021
Minimally invasive, indirect corneal neurotization using an ipsilateral sural nerve graft for early neurotrophic keratopathy	Lee BWH, Khan MA, Ngo QD, Tumuluri K, Samarawickrama C	*Am J Ophthalmol Case Rep*	2022
Corneal neurotization from the supratrochlear nerve with sural nerve grafts: a minimally invasive approach	Bains RD, Elbaz U, Zuker RM, Ali A, Borschel GH	*Plast Reconstr Surg*	2015
Direct Neurotization: Past, Present, and Future Considerations	Horen SR, Hamidian Jahromi A, Konofaos P	*Ann Plast Surg*	2022
Corneal neurotization	Koaik M, Baig K	*Curr Opin Ophthalmol*	2019
Corneal Neurotization-Indications, Surgical Techniques and Outcomes	Dragnea DC, Krolo I, Koppen C, Faris C, Van den Bogerd B, Ní Dhubhghaill S	*J Clin Med*	2023
Corneal neurotization in the management of neurotrophic keratopathy: A review of the literature	Saad S, Labani S, Goemaere I, Cuyaubere R, Borderie M, Borderie V, Benkhatar H, Bouheraoua N	*J Fr Ophtalmol*	2023
Korneale Neurotisation	Lueke JN, Holtmann C, Beseoglu K, Geerling G	*Ophthalmologe*	2020
Corneal neurotization for neurotrophic keratopathy: Review of surgical techniques and outcomes	Liu CY, Arteaga AC, Fung SE, Cortina MS, Leyngold IM, Aakalu VK	*Ocul Surf*	2021
Corneal Neurotization: Review of a New Surgical Approach and Its Developments	Wolkow N, Habib LA, Yoon MK, Freitag SK	*Semin Ophthalmol*	2019
Seeing through the evidence for corneal neurotization	Jowett N, Pineda R 2nd	*Curr Opin Otolaryngol Head Neck Surg*	2021
Bilateral Corneal Neurotization for Ramos-Arroyo Syndrome and Developmental Neurotrophic Keratopathy: Case Report and Literature Review	Rowe LW, Berns J, Boente CS, Borschel GH	*Cornea*	2023
Corneal Neurotization: Preoperative Patient Workup and Surgical Decision-making	Daeschler SC, Woo JH, Hussein I, Ali A, Borschel GH	*Plast Reconstr Surg Glob Open*	2023
Clinical outcomes of corneal neurotization using sural nerve graft in neurotrophic keratopathy	Saini M, Kalia A, Jain AK, Gaba S, Malhotra C, Gupta A, Soni T, Saini K, Gupta PC, Singh M	*PLoS One*	2023
Corneal Neurotization Using the Great Auricular Nerve for Bilateral Congenital Trigeminal Anesthesia	Lau N, Osborne SF, Vasquez-Perez A, Wilde CL, Manisali M, Jayaram R	*Cornea*	2022
Herpetic Corneal Keratopathy Management Using Ipsilateral Supratrochlear Nerve Transfer for Corneal Neurotization	Lin CH, Lai LJ	*Ann Plast Surg*	2019
Minimally-Invasive Corneal Neurotization (MICN): 10 Year Update in Technique and Lessons Learned Including Novel Donor Transfer of the Great Auricular Nerve	Gross JN, Bhagat N, Tran K, Liu S, Boente CS, Ali A, Borschel GH	*Plast Reconstr Surg*	2023
Minimally Invasive Corneal Neurotization With Acellular Nerve Allograft: Surgical Technique and Clinical Outcomes	Leyngold IM, Yen MT, Tian J, Leyngold MM, Vora GK, Weller C	*Ophthalmic Plast Reconstr Surg*	2019
Endoscopic Corneal Neurotization: Technique and Initial Experience	Leyngold I, Weller C, Leyngold M, Tabor M	*Ophthalmic Plast Reconstr Surg*	2018
Neurotization of the human cornea—A comprehensive review and an interim report	Rathi A, Bothra N, Priyadarshini SR, Achanta DSR, Fernandes M, Murthy SI, Kapoor AG, Dave TV, Rath S, Yellinedi R, Nuvvula R, Dendukuri G, Naik MN, Ramappa M	*Indian J Ophthalmol*	2022
Acellular nerve allografts in corneal neurotisation: an inappropriate choice	Jowett N, Pineda Ii R	*Br J Ophthalmol*	2020
The supraorbital and supratrochlear nerves for ipsilateral corneal neurotization: anatomical study	Kikuta S, Yalcin B, Iwanaga J, Watanabe K, Kusukawa J, Tubbs RS	*Anat Cell Biol*	2020
Neurotrophic Keratitis in a Pediatric Patient With Goldenhar Syndrome and Trigeminal Aplasia Successfully Treated by Corneal Neurotization	Rollon-Mayordomo A, Mataix-Albert B, Espejo-Arjona F, Herce-Lopez J, Lledo-Villar L, Caparros-Escudero C, Infante-Cossio P	*Ophthalmic Plast Reconstr Surg*	2022
Anatomic characteristics of supraorbital and supratrochlear nerves relevant to their use in corneal neurotization	Domeshek LF, Hunter DA, Santosa K, Couch SM, Ali A, Borschel GH, Zuker RM, Snyder-Warwick AK	*Eye (Lond)*	2019
The Second Division of Trigeminal Nerve for Corneal Neurotization: A Novel One-Stage Technique in Combination With Facial Reanimation	Gennaro P, Gabriele G, Aboh IV, Cascino F, Menicacci C, Mazzotta C, Bagaglia S	*J Craniofac Surg*	2019
Lateral Antebrachial Cutaneous Nerve as Autologous Graft for Mini-Invasive Corneal Neurotization (MICORNE)	Bourcier T, Henrat C, Heitz A, Kremer SF, Labetoulle M, Liverneaux P	*Cornea*	2019
Corneal neurotisation by great auricular nerve transfer and scleral-corneal tunnel incisions for neurotrophic keratopathy	Jowett N, Pineda Ii R	*Br J Ophthalmol*	2019
Corneal neurotization with a great auricular nerve graft: effective reinnervation demonstrated by in vivo confocal microscopy	Benkhatar H, Levy O, Goemaere I, Borderie V, Laroche L, Bouheraoua N	*Cornea*	2018
Outcomes of corneal neurotisation using processed nerve allografts: a multicentre case series	Sweeney AR, Wang M, Weller CL, Burkat C, Kossler AL, Lee BW, Yen MT	*Br J Ophthalmol*	2022

**Table 2 jcm-13-02268-t002:** The most relevant techniques described in the literature are reported and summarized herein. The “Type of coaptation” column indicates the type of nerve suture between the donor and the graft. This can be an end to end (E-E), end to side (E-S), or side to side (S-S).

Authors	Year	Type of Study	Patients	Eyes	Mean Age	Direct	Indirect	Donor Nerve	Nerve Graft	Type of Coaptation	Technique
Benkhatar H, Levy O, Goemaere I, Borderie V, Laroche L, Bouheraoua N.	2018	Case Report	1	1	58	✓	-	Ipsilateral Great Auricular	-	-	Open
Charlson ES, Pepper JP, Kossler AL.	2022	Case Report	1	1		-	✓	Contralateral Supraorbital + Supratrochlear	Dual Sural	E-E	Open
Giannaccare G, Bolognesi F, Biglioli F, Marchetti C, Mariani S, Weiss JS, Allevi F, Cazzola FE, Ponzin D, Lozza A, Bovone C, Scorcia V, Busin M, Campos EC	2020	Case Series	3	3	61	✓	-	Contralateral Supraorbital and/or Contralateral Supratrochlear	-	-	Open
Rusňák Š, Hecová L, Štěpánek D, Sobotová M.	2021	Case Report	1	1	22	-	✓	Contralateral Supraorbital	Sural	E-E	Open
Lee BWH, Khan MA, Ngo QD, Tumuluri K, Samarawickrama C.	2022	Case Report	1	1	11	-	✓	Ipsilateral Supratrochlear	Sural	E-E	Open
Rowe LW, Berns J, Boente CS, Borschel GH.	2023	Case Report + Literarature Review	1	2	17	-	✓	Great Auricular	Sural	E-E	Open
Lau N, Osborne SF, Vasquez-Perez A, Wilde CL, Manisali M, Jayaram R.	2022	Case Report	1	2	4	-	✓	Great Auricular	Sural	E-E	Open
Lin CH, Lai LJ.	2019	Case Series	13	13	61,8	✓	-	Ipsilateral Supratrochlear	-	-	Open
Leyngold IM, Yen MT, Tian J, Leyngold MM, Vora GK, Weller C.	2019	Case Series	7	7	46	-	✓	Supratrochlear (5) Supraorbital (1) Infraorbital (1)	Acellular Allograft	E-E E-E E-S	Open
Leyngold I, Weller C, Leyngold M, Tabor M.	2018	Case Report	1	1	83	✓	-	Contralateral Supraorbital	-	-	Endoscopic
Rollon-Mayordomo A, Mataix-Albert B, Espejo-Arjona F, Herce-Lopez J, Lledo-Villar L, Caparros-Escudero C, Infante-Cossio P.	2022	Case Report	1	1	6	-	✓	Contralateral Supratrochlear	Sural	E-E	Open
Bourcier T, Henrat C, Heitz A, Kremer SF, Labetoulle M, Liverneaux P.	2019	Case Report	1	1	32	-	✓	Contralateral Supraorbital	Lateral Antebrachial Cutaneous	E-E	Open
Sweeney AR, Wang M, Weller CL, Burkat C, Kossler AL, Lee BW, Yen MT.	2022	Case Series	17	17	42,6	-	✓	Ipsilateral or Contralateral Supratrochlear or Supraorbital	Acellular Allograft	E-E	Open
Gennaro P, Gabriele G, Aboh IV, Cascino F, Menicacci C, Mazzotta C, Bagaglia S.	2019	Case Report	1	1	58	✓	-	Ipsilateral Infraorbital	-	-	Open
Benkhatar H, Levy O, Goemaere I, Borderie V, Laroche L, Bouheraoua N.	2018	Case Report	1	1	58	-	✓	Supratrochlear	Great Auricular	E-E	Open

**Table 3 jcm-13-02268-t003:** Factors and variables determining the choice of the donor nerve.

Choice of the Donor Nerve: Variables and Factors to Be Considered.
Sensory integrity (peripheral/central trigeminal disease)
Nerve caliber
Axon count
Anatomical vicinity
NK severity
Previous surgeries
Unilateral or bilateral cornea anesthesia
Surgical accessibility
Surgeon’s expertise
Surgeon’s preference

**Table 4 jcm-13-02268-t004:** Positive and negative prognostic factors influencing surgical outcome of CN.

Factors Influencing Surgical Outcome	
Positive	Negative
B vitamins in the diet	Age
End to end coaptation	Time since denervation *
Caliber of donor nerve	Distance of graft coaptation
Autologous graft	Smoking
	Diabetes
	Allograft

* Although it does not influence directly the surgical outcomes, longer time since denervation is often linked to worse conditions of the affected cornea.

## Data Availability

Not applicable.

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
