# Peer review of "Insights on the Choice and Preparation of the Donor Nerve in Corneal Neurotization for Neurotrophic Keratopathy: A Narrative Review"

_jcm, 2024, doi:10.3390/jcm13082268_

Round 1

Reviewer 1 Report

Comments and Suggestions for Authors

Dear Authors, congratulations on Your interesting article.
The article presents the problem of neurotrophic keratopathy in an accessible way. The corneal neurotization or nerve coaptation in neurotrophic keratopathy is not often discussed, which makes the article even more interesting. A historical outline of surgical methods constituting the basis for corneal neurotization is briefly presented and the concepts of direct and indirect neurotization are explained. Both tables included in the text are described in a clear way.
The results section of the article describes the innervation of the cornea in an accessible way. Next, the authors describe the types of treatments and list the factors that positively and unfavorably influence the treatment results. Nerves used in direct and indirect neurotization treatments are presented. This part requires a lot of concentration from the reader, among other things, due to numerous abbreviations.
Then, the representative clinical case is provided, focusing on the course of the operation itself. In the last part of the article other aspects of the problem of corneal neurotization were discussed, primarily based on the results of a non-randomized multicentric interventional study by Fugagnolo and colleagues. In addition, the results of research on acellular nerve allografts were presented.

My comments are as follows:
-  In Table No. 2, it is worth adding an explanation of the "type of coaptation" column and the abbreviations it contains: E-E (end to end), E-S (end to side). There is another possible type of coaptation: side-to-side. Although, the above abbreviations are explained in the text of the article. 
- In lines 106 and 107: "Neurotrophic keratopathy is a degenerative disease that leads to progressive trigeminal damage resulting in corneal hypoesthesia" – reversal of causation. Progressive damage to the trigeminal nerve leads to neurotrophic keratopathy, which is characterized by corneal hypoesthesia.

- It would be helpful to present in a table the factors influencing negative and positive treatment outcomes

- lines 124 - 127 describe better results using nerve coaptation type end to end and the characteristics of the advantage of this solution (better functional recovery, associated with increased nerve fiber count, area, and density). Are the Authors able to present the data statistically?

- Many abbreviations have been given for the nerves used in direct and indirect neurotization of the cornea. Presenting the nerves and their abbreviated names in a table with their assignment to the type of neurotization (direct/direct/both) and the length of the graft (or the preferred length of the graft for a given nerve) would definitely facilitate the acquisition of information
- The part of the work containing the case description is very interesting. It does not include the results of surgical treatment. Do the Authors know the patient's follow-up after surgery and what was the cause of the patient's neurotrophic keratopathy?

- The discussion takes into account many variables obtained in the cited studies. Again, do the authors consider capturing data in tabular form?

The presented article is very valuable. The content is challenging for the reader. The use of schematic drawings and tables may facilitate the transfer of information.

Author Response

Dear Reviewer,
Please find attached our reply to your valuable advice,
Best regards

Reviewer 2 Report

Comments and Suggestions for Authors

Neurotrophic keratopathy (NK) is a sight threating condition resulting from corneal denervation due to various causes of trigeminal nerve dysfunctions. Surgical techniques for corneal neurotization (SCN) aiming to restore corneal sensitivity, represents a promising avenue in treating of NK, offering tailored approaches based on patient history and surgical expertise. This manuscript reviewed the new progress in this field, which is of importance for reference to improve the clinical selection of SCN  strategy. Some issues should be clarified before publication:

1. SCN can be performed through a direct approach directly suturing a sensitive nerve to the affected cornea or indirectly through a nerve auto/allograft.  Dese the sensitive nerve here refered to trigeminal nerve or including other ones?

2. CN represents a promising avenue in treating of NK.  The mean of CN should be interpreted at the first time it appears in the manuscript.

3.  Acellular nerve allografts represent a promising element in the armamentarium of the multidisciplinary team facing this disease and deserve basic science refinements and future clinical trials. What is the difference between Acellular nerve allografts and traditional nerve allografts?

Author Response

(The authors gave the same response as above.)
